# The Kinetics of Sorption–Desorption Phenomena: Local and Non-Local Kinetic Equations

**DOI:** 10.3390/molecules27217601

**Published:** 2022-11-05

**Authors:** Giovanni Barbero, Antonio M. Scarfone, Luiz R. Evangelista

**Affiliations:** 1Dipartimento di Scienza Applicata e Tecnologia, Politecnico di Torino, Corso Duca degli Abruzzi 24, 10129 Torino, Italy; 2Istituto dei Sistemi Complessi, Consiglio Nazionale delle Ricerche (ISC-CNR), c/o Politecnico di Torino, Corso Duca degli Abruzzi 24, 10129 Torino, Italy; 3Moscow Engineering Physics Institute, National Research Nuclear University MEPhI, Kashirskoye Shosse 31, 115409 Moscow, Russia; 4Departamento de Física, Universidade Estadual de Maringá, Avenida Colombo, 5790, Maringá 87020-900, PR, Brazil

**Keywords:** Langmuir equation, local and non-local kinetic equation, sorption–desorption phenomena

## Abstract

The kinetics of adsorption phenomena are investigated in terms of local and non-local kinetic equations of the Langmuir type. The sample is assumed in the shape of a slab, limited by two homogeneous planar-parallel surfaces, in such a manner that the problem can be considered one-dimensional. The local kinetic equations in time are analyzed when both saturation and non-saturation regimes are considered. These effects result from an extra dependence of the adsorption coefficient on the density of adsorbed particles, which implies the consideration of nonlinear balance equations. Non-local kinetic equations, arising from the existence of a time delay characterizing a type of reaction occurring between a bulk particle and the surface, are analyzed and show the existence of adsorption effects accompanied by temporal oscillations.

## 1. Introduction

When a solid surface comes in contact with a liquid in which impurities are dissolved, according to the interaction energy between these impurities and the surface, they can either be attracted or repelled from the surface [1,2]. There is also the possibility that will be handled here in which the particles are attracted to the surface and remain stuck on it. In the equilibrium state, the number of particles passing from the liquid to the surface (adsorbed) is equal to that passing from the surface to the liquid (desorbed). A simple description of the phenomenon, in cases where the particles are neutral and long-range effects can be neglected, is based on the kinetic equation relating the rate of adsorbed particles with the adsorption phenomenon of particles from the bulk, and the desorption of adsorbed particles from the surface [3]. In the approximation whereby the adsorption energy is independent of the surface density of adsorbed particles and all adsorbing sites are equivalent [4], the kinetic equation reads
(1)$$\frac{d\sigma (t)}{dt}=k\;n\left(t\right)-\frac{1}{\tau }\;\sigma \left(t\right)\;,$$
where σ(t) is the surface density of adsorbed particles, n(t) is the bulk density of adsorbable particles, *k* and τ are the adsorption coefficient and the desorption time, respectively. Expression (Equation 1) has been widely used to analyze the time-dependence of the surface density of adsorbed particles under different conditions [5,6,7,8]. It has been also employed in the investigation of the adsorption phenomenon in the presence of an external electric field [9,10,11]. A generalization of Equation (Equation 1) is discussed in [12]. The adsorption parameters depend, clearly, from the adsorbing surface and the adsorbable particles. They depend also on the temperature and on the presence of an external electric field, as reported in Refs. [7,10,12,13,14,15,16].

Indicating the surface density of adsorbing sites by $\sigma_{0}$, in the case of $\sigma \ll \sigma_{0}$, the adsorption coefficient is independent of σ [17]. On the contrary, for $\sigma ∼\sigma_{0}$, the possibility of a multilayer adsorption has to be considered. In conditions where only the formation of a monolayer is possible, k→k(σ) has to be such that for $\sigma \geq \sigma_{0}$, k(σ)<0, indicating that when the surface density overcomes the density of sites, the adsorption phenomenon is no longer possible. A proposed form for the dependence of k(σ) on σ(t) is [18]
(2)$$k\left(\sigma \right)=k\;\left[1-{\left(\frac{\sigma }{\sigma_{0}}\right)}^{p}\right]\;,$$
with p>1. When the formation of a multilayer is possible, k(σ) and τ can depend on the number of layers already adsorbed.

The work focuses on the kinetic evolution of the system towards the equilibrium state, where n(t) and σ(t) are time-independent. To solve the problem, in addition to the kinetic balance equation, one has to also consider conservation of the total particle number comprising the bulk plus the surface system.

The analysis deals with a sample limited by two homogeneous planar-parallel surfaces at a distance *d* apart. Thus, the bulk density of particles depends only on the distance of the considered point from the limiting surfaces, i.e., the mathematical problem is one-dimensional. This is the so-called slab approximation, which is assumed henceforth. It is also assumed that the surface forces responsible for the adsorption phenomenon are short range, in such a manner that in the bulk n(t) can be considered uniform. This approximation is valid if the diffusion time $\tau_{D}$ is extremely small with respect to the characteristic times of the adsorption phenomenon. Indicating the diffusion coefficient by *D* of the adsorbable particles, $\tau_{D}=d^{2}/D$. If $d∼10^{-5}\;$m and $D∼10^{-9}\;$m${}^{2}$/s, $\tau_{D}∼0.1\;$s. The analysis presented in the following, works well for samples of few μm in thickness, and for ordinary diffusion coefficient, such as Na${}^{+}$ in water [3]. In cases where the adsorbing surface is in contact with a reservoir, fixing the value of adsorbable particles, the quantity n(t) appearing in the kinetic equation, with or without saturation, coincides with $n_{0}$, the bulk equilibrium density. In this case, since d→∞, it is no longer necessary to impose conservation of the number of particles. Several measurements reported in the literature correspond to this situation, and our analysis can be easily generalized and applied to these cases.

The goal of the paper is to develop a model capable of analyzing interesting data reported in the literature [5,6,7,8,9,10,11,12,13,14,15,16] relevant to the dependence on surface density of the adsorbed particles versus the concentration and the time evolution of the surface density. Furthermore, a model is proposed that is able to explain the time-oscillating behavior of the surface density of adsorbed particles [19,20,21,22,23,24,25,26] toward the equilibrium state.

The paper is organized as follows. Section 2 is dedicated to the investigation of local kinetic equations from three different perspectives. In Section 2.1, the dynamical evolution of the surface density of adsorbed particles when the kinetic equation is well described by Langmuir isotherm is analyzed. This limiting case works well when the surface density of adsorbed particles in the steady state is extremely small with respect to the surface density of adsorbing sites. The scenario for large adsorption, when saturation effects are important, is considered in Section 2.2, where the adsorption coefficient depends linearly on the surface density of adsorbed particles (p=1) is investigated first. The relevant problem for which the dependence is quadratic (p=2) is discussed in Section 2.3. Section 3 deals with non-local kinetic equations involving a time delay. The analysis shows that the adsorption–desorption process could present a combination of saturation accompanied by an oscillating behavior, when the time delay is comparable with the desorption time. Lastly, the conclusions are provided in Section 4.

## 2. Local Kinetic Equations

In this section, the adsorption–desorption phenomena are analyzed in the presence of local kinetic equation when saturation and non-saturation regimes are considered. These equations are local in time because the reaction characterizing the adsorption–desorption process at the surfaces are assumed to be instantaneous, i.e., there is no time delay characterizing the bulk-surface dynamics.

### 2.1. No Saturation: k(σ)=k

Let us consider first the simple case in which the kinetic equation describing the adsorption phenomenon (Equation 1) involves an adsorption term proportional to the bulk density of adsorbable particles n(t), related to the constant adsorption coefficient *k*, and a desorption term proportional to the surface density of adsorbed particles, related to the desorption time τ. Assuming a uniform bulk distribution of the adsorbable particles n(t), the conservation of the number of particles (bulk plus surface) implies that
(3)$$n\left(t\right)\;d+2\;\sigma \left(t\right)=n_{0}\;d\;.$$

For the analysis presented in the following, it is useful to measure the time in units of τ, i.e., T=t/τ, and to introduce the dimensionless parameter defined as $u=\tau /\tau_{\kappa }=2\;k\;\tau /d$, where kτ is an intrinsic length and $\tau_{\kappa }=d/2\;\kappa$ is an intrinsic time, both connected to the adsorption phenomenon [17], and *d* is the thickness of the sample. The maximum surface density of adsorbed particles is $\sigma_{M}=n_{0}\;d/2$. For this reason, in the following, the surface and the bulk densities of particles will be measured in units of $\sigma_{M}$ and $n_{0}$, i.e.,
(4)$$S=\frac{\sigma }{\sigma_{M}}\;,\;\mathrm{and}\;N=\frac{n}{n_{0}}\;,$$
in such a way that Equations (Equation 1) and (Equation 3) can be rewritten as
(5)$$\frac{dS}{dT}=u\;N-S\;,$$
and
(6)N+S=1,
respectively. Substitution of Equation (Equation 6) into Equation (Equation 5) yields
(7)$$\frac{dS}{dT}+\left(1+u\right)\;S=u\;.$$

In the equilibrium state, dS/dT=0. It follows that in the steady state the dimensionless surface density of adsorbed particles and bulk density of adsorbable particles are given, respectively, by
(8)$$S^{*}=\frac{u}{1+u}\;,\;\mathrm{and}\;N^{*}=\frac{1}{1+u}\;.$$

In absolute units, the corresponding surface and bulk particles densities are
(9)$$\sigma^{*}=\frac{u}{1+u}\;\sigma_{M}\;,\;\mathrm{and}\;n^{*}=\frac{1}{1+u}\;n_{0}\;.$$

The kinetic evolution of S(T) is obtained by solving the differential Equation (Equation 7) with the boundary condition S(0)=0. Simple calculations give
(10)$$S\left(T\right)=\frac{u}{1+u}\;\left[1-e^{-(1+u)\;T}\right]\;.$$

The effective relaxation time, $\tau_{R}$, of the surface density of adsorbed particles σ(t), in absolute units, is
(11)$$\tau_{R}=\frac{\tau }{1+u}=\frac{\tau \;\tau_{\kappa }}{\tau +\tau_{\kappa }}\;.$$

Note that Equation (Equation 11) can be rewritten as
(12)$$\frac{1}{\tau_{R}}=\frac{1}{\tau }+\frac{1}{\tau_{\kappa }}\;,$$
indicating that, in the case in which τ is very different from $\tau_{\kappa }$, $\tau_{R}$ is approximatively equal to the smallest between $\tau_{\kappa }$ and τ. When u→0, i.e., d→∞, $\tau_{R}\to \tau$. This case represents a surface in contact with a reservoir of particles. The opposite case, in which u→∞, i.e., d→0, yields $\tau_{R}\to 0$, which is not of experimental interest.

### 2.2. Linear Saturation: p=1

A simple generalization of Equation (Equation 1) to take the saturation into account is obtained when one considers k(σ) in Equation (Equation 2), for p=1, namely:(13)$$\begin{array}{c} \frac{d\sigma (t)}{dt}=k\left(1-\frac{\sigma (t)}{\sigma_{0}}\right)\;n\left(t\right)-\frac{1}{\tau }\;\sigma \left(t\right)\;, \end{array}$$
where $\sigma_{0}$ is the surface density of adsorbing sites. In terms of the dimensionless quantities defined in (Equation 4), Equation (Equation 13) can be rewritten as
(14)$$\frac{dS}{dT}=u\left(1-r\;S\right)\;N-S\;,$$
where
(15)$$r=\frac{\sigma_{M}}{\sigma_{0}}=\frac{n_{0}\;d}{2\;\sigma_{0}}\;,$$
is a dimensionless parameter comparing the maximum density of adsorbable particles with the density of sites on the surface. Note that for a fixed adsorbing surface in contact with a solution the parameter $\sigma_{0}$ is fixed and different values of *r* correspond to different bulk equilibrium concentrations of adsorbable particles. In the present case, the conservation of particles is still represented by Equation (Equation 6). The saturation is important only in the case where r>1. When r→0, the analysis presented above has to be recovered.

Substitution of Equation (Equation 6) into Equation (Equation 14) yields for S(T) the ordinary differential equation:(16)$$\frac{dS}{dT}=u\;r\;S^{2}-\left(1+u+u\;r\right)\;S+u\;.$$

As before, in the equilibrium state dS/dT=0, and the surface density characterizing the steady state is the solution of the second-order equation
(17)$$u\;r\;S^{2}-\left(1+u+u\;r\right)\;S+u=0\;,$$
whose solutions are
(18)$$S_{1}=\frac{1+u+r\;u-\sqrt{{\left(1+u+r\;u\right)}^{2}-4\;r\;u^{2}}}{2\;r\;u}\;,$$
and
(19)$$S_{2}=\frac{1+u+r\;u+\sqrt{{\left(1+u+r\;u\right)}^{2}-4\;r\;u^{2}}}{2\;r\;u}\;.$$
Since
(20)$${\left(1+u+r\;u\right)}^{2}-4\;r\;u^{2}=\left[1+{\left(1-\sqrt{r}\right)}^{2}\;u\right]\;\left[1+{\left(1+\sqrt{r}\right)}^{2}\;u\right]>0\;,$$
for all meaningful physical parameters, $S_{1}$ and $S_{2}$ are real quantities. From Equations (Equation 18) and (Equation 19), in the limit of r→0, one obtains
(21)$$S_{1}=\frac{u}{1+u}-\frac{u^{2}}{{\left(1+u\right)}^{3}}\;r+O\left(r^{2}\right)\;,$$
and
(22)$$S_{2}=\frac{1+u}{ru}+\frac{1}{1+u}+\frac{u^{2}}{{\left(1+u\right)}^{3}}r+O\left(r^{2}\right)\;.$$

Since, in the r→0 limit, one has to recover the result given by (Equation 8), it is possible to conclude that the actual surface density in the equilibrium state is $S_{1}$. The dependence of the equilibrium surface density versus *r* is shown in Figure 1.

Notice that in the limit of r→∞, corresponding to a sample of infinite thickness, from Equation (Equation 18) one obtains
(23)$$S_{1}=\frac{1}{r}+O\left(r^{-2}\right)\;.$$

Taking into account that the actual surface density is given by $\sigma =\sigma_{M}\;S$, if one expresses σ in terms of $\sigma_{0}$, then one has $\sigma /\sigma_{0}=rS$. It follows that, from Equation (Equation 23), in the considered limit of large *r*
(24)$$\lim_{r\to \infty }\frac{\sigma_{1}}{\sigma_{0}}=1\;.$$
which states that, in an infinite sample, all adsorbing sites are occupied.

The evolution of S(T) toward to the equilibrium state is determined by solving Equation (Equation 16). A simple calculation gives
(25)$$S\left(T\right)=S_{1}\;S_{2}\;\frac{1-e^{-\alpha \;T}}{S_{2}-S_{1}e^{-\alpha \;T}}\;,$$
where
(26)$$\alpha =\sqrt{{\left(1+u+r\;u\right)}^{2}-4\;r\;u^{2}}>0,$$
as it follows from Equation (Equation 20). The time dependence of S(T) given by Equation (Equation 25) is shown in Figure 2 for two different values of *r*.

The time dependence of the dimensionless surface density of adsorbed particles compare qualitatively well with the experimental data reported in Refs. [8,10,14], relevant to the influence of the equilibrium concentration on the adsorption phenomenon. The relaxation time, in absolute units for σ(t), is then
(27)$$\tau_{R}=\frac{\tau }{\sqrt{{\left(1+u+r\;u\right)}^{2}-4\;r\;u^{2}}}\;.$$

One notices that the divergence of $\tau_{R}$ is only apparent because the quantity ${\left(1+u+r\;u\right)}^{2}-4\;r\;u^{2}$ is always positive for all values of *r* and *u* physically meaningful, as stressed above. From Equation (Equation 27), in the r→0 limit, one gets
(28)$$\tau_{R}=\frac{\tau }{1+u}\;\left[1-\frac{1-u}{{\left(1+u\right)}^{2}}\;u\;r\right]+O\left(r^{2}\right)\;,$$
that, in the same limit, coincides with (Equation 11), as expected. The dependence of $\tau_{R}$ on *r* is shown in Figure 3, where are also reported the limiting cases corresponding to small and large values of *r*.

### 2.3. Quadratic Saturation: p=2

Let us consider now the case in which the kinetic equation is represented by
(29)$$\frac{d\sigma (t)}{dt}=k\;\left[1-{\left(\frac{\sigma (t)}{\sigma_{0}}\right)}^{2}\right]\;n\left(t\right)-\frac{1}{\tau }\;\sigma \left(t\right)\;,$$
where, as before, $\sigma_{0}$ is the surface density of adsorbing sites. In terms of reduced quantities, it takes the form
(30)$$\frac{dS}{dT}=u\;\left(1-r^{2}\;S^{2}\right)\;N-S\;,$$
and the equation stating the conservation of the number of particles is still represented by Equation (Equation 6). Differential Equation (Equation 29), taking into account Equation (Equation 6), can be rewritten as
(31)$$\frac{dS}{dT}=u\;r^{2}\;S^{3}-u\;r^{2}\;S^{2}-\left(1+u\right)\;S+u\;.$$

In this way, the equilibrium value of *S* is obtained by solving the third-order equation
(32)$$\frac{dS}{dT}=0\;.$$

By inspection, one observes that for all physically meaningful values of *r* and *u*, Equation (Equation 32) always contains three distinct real solutions as shown in Figure 4. The searched solution to the kinetic Equation (Equation 29) is the one that, in the $\lim_{r\to 0}S^{*}$, reduces to Equation (Equation 8).

Indicating by $S_{1}$, $S_{2}$ and $S_{3}$ the three solutions of Equation (Equation 32), a simple calculation shows that in the r→0 limit they tend respectively to
(33)$$\begin{array}{ccc} S_{1} & \to & \sqrt{\frac{1+u}{u}}\;\;\frac{1}{r}+O\left(r^{-1}\right)\;, \end{array}$$
(34)$$\begin{array}{ccc} S_{2} & \to & -\sqrt{\frac{1+u}{u}}\;\;\frac{1}{r}+O\left(r^{-1}\right)\;, \end{array}$$
(35)$$\begin{array}{ccc} S_{3} & \to & \frac{u}{1+u}\left[1-\frac{u^{2}}{{\left(1+u\right)}^{3}}\;r^{2}\right]+O\left(r^{-1}\right)\;, \end{array}$$
from which one deduces that solutions $S_{3}$ tends to the value given by (Equation 8), as expected. In the opposite limit of r→∞, the solutions tend to
(36)$$\begin{array}{ccc} S_{1} & \to & 1+\frac{1}{u\;r^{2}}+O\left(r^{-2}\right)\;, \end{array}$$
(37)$$\begin{array}{ccc} S_{2} & \to & -\frac{1}{r}-\frac{1}{2\;r^{2}u}+O\left(r^{-2}\right)\;, \end{array}$$
(38)$$\begin{array}{ccc} S_{3} & \to & \frac{1}{r}-\frac{1}{2\;u\;r^{2}}+O\left(r^{-2}\right)\;, \end{array}$$
at the second order in 1/r. As already mentioned, the surface density in units of $\sigma_{0}$, are rS and tend to
(39)$$r\;S_{1}\to \infty ,\;r\;S_{2}\to -1,\;\mathrm{and}\;r\;S_{3}\to 1,$$
showing again that the solution of the problem also has the correct limit in the case of large *r*. The numerical solution of Equation (Equation 31) is shown in Figure 5, for u=0.1 and r=0.1, r=5, and r=10.

Finally, Figure 6 compares the time evolution of S(T) deduced in the case of no saturation (dash-dotted line), linear saturation (full line), and quadratic saturation (dashed line).

## 3. Non-Local Kinetic Equations

In the previous sections, a few kinetic equations describing the adsorption phenomenon, in the absence of saturation, Equation (Equation 1), in the presence of linear, Equation (Equation 13) and quadratic, Equation (Equation 29), saturation have been considered. These equations were solved by taking into account conservation of the number of particles, that we assumed as represented by Equation (Equation 3). This is a rather rough approximation, but allows one to derive some general properties on the adsorption. These kinetic equations are of the type
(40)$$\frac{d\sigma (t)}{dt}{|}_{t}=H\left[n\left(t\right),\sigma \left(t\right)\right]-\frac{1}{\tau }\;\sigma \left(t\right)\;,$$
i.e., they relate the time variation of σ(t) at *t* with the bulk density of adsorbable particles, and with the surface density of adsorbed particles, at the same time *t*. However, since the adsorption phenomenon implies a form of chemical reaction occurring between a bulk particle and the surface, the time rate dσ/dt at *t* depends on n(t) and σ(t) at an antecedent time t−δ, where δ depends on the particular reaction responsible for the adsorption. This particular time, δ, can differ from the desorption time, τ. In other words, usually, there is an intrinsic delay between dσ(t)/dt and, n(t) and σ(t). This delay is related to δ in such a way that Equation (Equation 40) has to be written as
(41)$$\frac{d\sigma (t)}{dt}{|}_{t+\delta }=H\left[n\left(t\right),\sigma \left(t\right)\right]-\frac{1}{\tau }\;\sigma \left(t\right)\;.$$

Assuming δ small with respect to the observation time *t*, from Equation (Equation 41) one gets
(42)$$\delta \;\frac{d^{2}\sigma \left(t\right)}{dt^{2}}+\frac{d\sigma (t)}{dt}=H\left[n\left(t\right),\sigma \left(t\right)\right]-\frac{1}{\tau }\;\sigma \left(t\right)\;,$$
i.e., now the kinetic equation contains a second order derivative in *t*.

To investigate the effect of the delay on the evolution of the surface density of adsorbed particles, Equation (Equation 1) is now generalized, with the condition (Equation 3). In the presence of delay, Equation (Equation 1) becomes
(43)$$\delta \;\frac{d^{2}\sigma \left(t\right)}{dt^{2}}+\frac{d\sigma (t)}{dt}=k\;n\left(t\right)-\frac{1}{\tau }\;\sigma \left(t\right)\;,$$
that, by taking into account Equation (Equation 3) and using dimensionless quantities, assumes the form:(44)$$\varepsilon \;\frac{d^{2}S}{dT^{2}}+\frac{dS}{dT}+\left(1+u\right)\;S=u\;,\;\mathrm{where}\;\varepsilon =\delta /\tau \;.$$

Equation (Equation 44) is the new version of Equation (Equation 7) in the presence of a time delay. The parameter ε compares the delay time δ with the desorption time τ.

To solve Equation (Equation 44), one has to impose the initial conditions on dS/dT and *S*. It will be assumed, as before, that the adsorption phenomenon begins at t=0, and hence S(0)=0 and, furthermore, that ${\left(dS/dT\right)|}_{t=0}=0$. In this framework, as it is evident from (Equation 44), the case ε=0 is singular, because the differential equation passes from the second to the first order. Solution of Equation (Equation 44) with the mentioned initial conditions, when 1−4(1+u)ε≠0, is
(45)$$S\left(T\right)=\frac{u}{1+u}\left[1-\frac{1}{\alpha_{2}-\alpha_{1}}\left(\alpha_{2}\;e^{\alpha_{1}\;T}-\alpha_{1}\;e^{\alpha_{2}\;T}\right)\right]\;,$$
where
(46)$$\begin{array}{ccc} \alpha_{1} & = & -\frac{1}{2\;\varepsilon }\left[1-\sqrt{1-4\;(1+u)\;\varepsilon }\right]\;, \end{array}$$
(47)$$\begin{array}{ccc} \alpha_{2} & = & -\frac{1}{2\;\varepsilon }\left[1+\sqrt{1-4\;(1+u)\;\varepsilon }\right]\;. \end{array}$$

In the particular case in which 1−4(1+u)ε=0, solution of Equation (Equation 44) with the above mentioned boundary conditions is
(48)$$S\left(T\right)=\frac{u}{1+u}\left[1-\left(1+\frac{T}{2\;\varepsilon }\right)\;e^{-T/(2\;\varepsilon )}\right]\;.$$

The exponents $\alpha_{1}$ and $\alpha_{2}$ depend critically on 1−4(1+u)ε. If $\varepsilon <\varepsilon_{c}=1/\left[4\;\left(1+u\right)\right]$, then $\alpha_{1}$ and $\alpha_{2}$ are negative real numbers, and the solution S(T) tends uniformly to u/(1+u). In this case, the relaxation toward to the equilibrium value of the surface density of adsorbed particles is characterized by two different relaxation times and the explicit solution is
(49)$$S\left(T\right)=\frac{u}{1+u}\left\{1-\frac{1}{\alpha }\;\left[\sinh\left(\frac{\alpha }{2\;\varepsilon }\;T\right)+\alpha \;\cosh\left(\frac{\alpha }{2\;\varepsilon }\;T\right)\right]\;e^{-T/(2\;\varepsilon )}\right\},$$
where $\alpha =\sqrt{4\;(1+u)\;\varepsilon -1}$. An expression of the type given by Equation (Equation 45) has been proposed to interpret the adsorption data by Vanegas et al. [27], and used more recently to analyze adsorption data in [15,16].

In the opposite case, where $\varepsilon >\varepsilon_{c}$, $\alpha_{1}$ and $\alpha_{2}$ are complex conjugate numbers, with a negative real part. The solution S(T) tends to u/(1+u) oscillating, and the explicit solution is
(50)$$S\left(T\right)=\frac{u}{1+u}\left\{1-\frac{1}{\alpha }\left[\sin\left(\frac{\alpha }{2\;\varepsilon }\;T\right)+\alpha \;\cos\left(\frac{\alpha }{2\;\varepsilon }\;T\right)\right]\;e^{-T/(2\varepsilon )}\right\}\;,$$
where $\alpha =\sqrt{1-4\;(1+u)\;\varepsilon }$.

Figure 7 shows the time evolution of *S* for u=0.1 and $\varepsilon =5\;\varepsilon_{c}$ (dash-dotted line), $\varepsilon =0.9\;\varepsilon_{c}$ (full line), and the solution for ε=0, i.e., assuming no delay (dashed line). As can be observed, all curves tend to the same limit u/(1+u). The inset of Figure 7 shows the evolution of *S* for small *T*. From this frame, it is clear that the case corresponding to δ=0 does not satisfy the initial condition on dS/dT=0.

The simple analysis presented above to take into account the delay δ was possible because the governing differential equation is linear. The same generalization in the case where the adsorption phenomenon is described by Equation (Equation 16) or by Equation (Equation 31) gives rise to the differential equations
(51)$$\begin{array}{ccc} & & \varepsilon \;\frac{d^{2}S}{dT^{2}}+\frac{dS}{dT}=u\;r\;S^{2}-\left(1+u+u\;r\right)\;S+u\;, \end{array}$$
(52)$$\begin{array}{ccc} & & \varepsilon \;\frac{d^{2}S}{dT^{2}}+\frac{dS}{dT}=u\;r^{2}\;S^{3}-u\;r^{2}\;S^{2}-\left(1+u\right)\;S+u\;, \end{array}$$
respectively. In presence of the saturation effect, it is no longer possible to obtain an analytical solution, because Equations (Equation 51) and (Equation 52) are no longer linear. However, a numerical solution can be easily obtained. Figure 8 compares the time evolution of *S* described by Equation (Equation 51), (dashed line), and by Equation (Equation 52), (dash-dotted line), for $\varepsilon =5\;\varepsilon_{c}$ with u=0.1 and r=100. Furthermore, in this case, the solutions are oscillating around the stable limiting value.

This oscillating behavior can be qualitatively compared with the ones exhibited by some acid monolayers whose adsorption is governed by physical forces, but in which the role of the head group has to be taken into account [19,20]. As pointed out in Ref. [19], an oscillating behavior such as the ones depicted in Figure 7 and Figure 8 is in agreement with the behavior found by Fourier transform IR spectroscopy for siloxane polymers chemisorbing to aluminia [20,21]. In these systems, measurements of the variation of physically adsorbed and chemically adsorbed segments show that the physisorption process is strong in the first instants, presenting a pronounced maximum and tending to a small value; the chemisorption process, on the contrary, tends to a saturation value. When these effects are combined, because they occur during the same process, the resulting curve may present oscillating behavior. The effect of the non-local term, represented by the second order time-derivative, may be connected with memory effects. Indeed, it may indicate that the adsorption process is much better described if one considers the memory of the preceding state for the molecules being adsorbed at the surface.

An effect of this kind is not unusual and can even be expected because, for instance, in the adsorption phenomenon by a solid surface, the collision of a molecule can be represented by different processes [22]. One of these is an elastic scattering, that occurs when there is no loss in translational energy during the collision; however, if the molecule is still in a weakly bound state, even if it is on the surface, the thermal motion of the surface atoms can cause the molecule to desorb and this represents another kind of process. Finally, when the molecule collides with a surface, it loses energy and is converted into a state where it remains on the surface for a reasonable length of time [23]. These phenomena indicate that the actual position of the molecule on the surface has a memory of its incoming state, eventually modifying the adsorption–desorption rates. An alternative analysis of this process has been carried out by considering modified linear kinetic equation in which some suitable choices for a kernel in the desorption rate accounted for the importance of the physisorption or of the chemisorption, according to the time scale governing the adsorption phenomena [19]. In the present analysis, the oscillating behavior is obtained with linear and nonlinear kinetic equations incorporating an expected time delay in the reaction of the molecules from the bulk at the surface, during the adsorption process and may be relevant to understand oscillation phenomena in polymers [24], multicomponent adsorption [25], surfactant ions [26], among many others.

## 4. Conclusions

The work approaches the kinetic equation in the Langmuir approximation from two different perspectives. The first deals with a balance equation that is local in time, but in which a phenomenological adsorption coefficient intervenes that may depend on the density of adsorbed particles according to a general power law. After formulating the problem in more general terms, the conventional case k(σ)=const. is considered and two other possibilities: a linear and a quadratic dependence. In such cases, the density of particles at equilibrium is described by algebraic equations and the time evolution by nonlinear differential equations. Thus, it is possible to analyze various adsorption regimes that are essentially governed by a saturation phenomenon dictated by the finite number of sites at the interface. In the second perspective, the adsorption–desorption phenomena are analyzed in a non-local picture in time, that is, it is assumed that there exists a time delay between the arrival of the particle in the bulk, but in the vicinity of the interface, and the adsorption–desorption itself, taking place over the interface. In this case, the entire kinetics are described by an equation involving a second derivative in time, which reduces to first-order when the time delay is negligible. In this context, it is found that the adsorption–desorption regimes could be accompanied by an oscillation phenomenon in the coverage ratio behavior. A similar phenomenon can be found using linear kinetic equations, but containing temporal kernels that take memory effects into account. There is, therefore, a very complete approach to the adsorption–desorption phenomena of neutral particles (although the extension to the case of charged particles can also be considered from the same perspective) at the liquid (bulk)–solid (surface) interface.

## Figures and Tables

**Figure 1 molecules-27-07601-f001:**
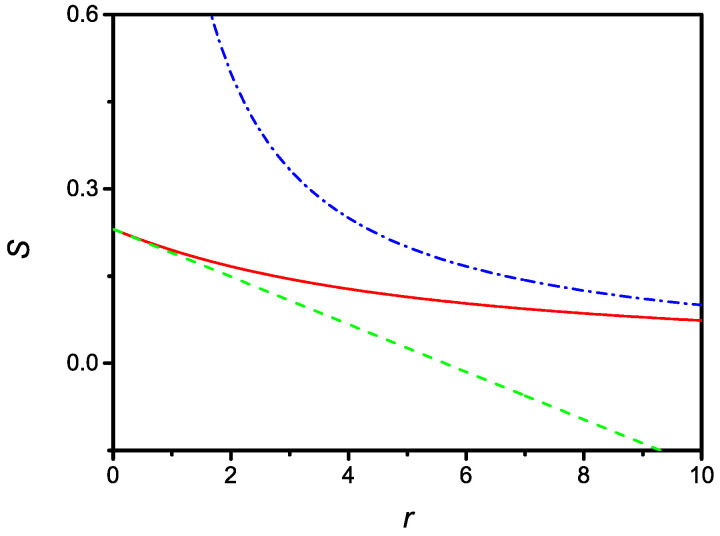
Dependence of the dimensionless surface density of the steady state versus the dimensionless parameter *r*. Dashed and dash-dotted lines represent the limiting cases corresponding to small and large *r*, respectively, as given by Equations (Equation 21) and (Equation 23). The curves are drawn for u=0.3 or $\tau =0.3\;\tau_{\kappa }$, i.e., the adsorption time is approximately three-times faster than the desorption time.

**Figure 2 molecules-27-07601-f002:**
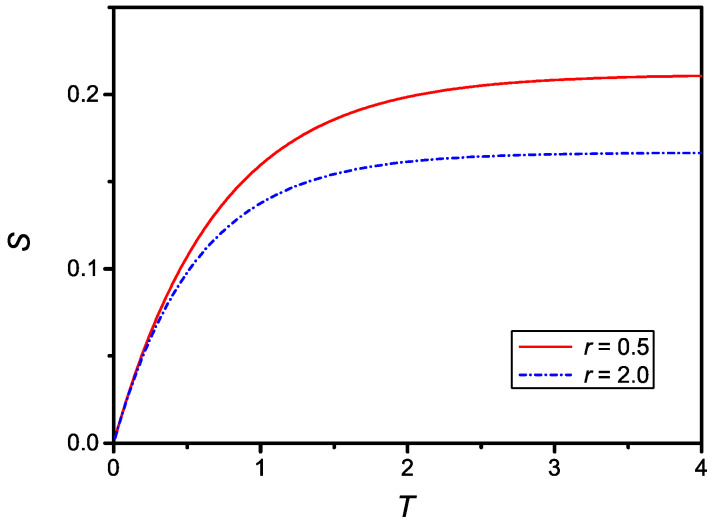
Dependence on the dimensionless surface density of adsorbed particles, S(T), versus the dimensionless time, *T*, for u=0.3, r=0.5, (dash-dotted line) and r=2, (full line).

**Figure 3 molecules-27-07601-f003:**
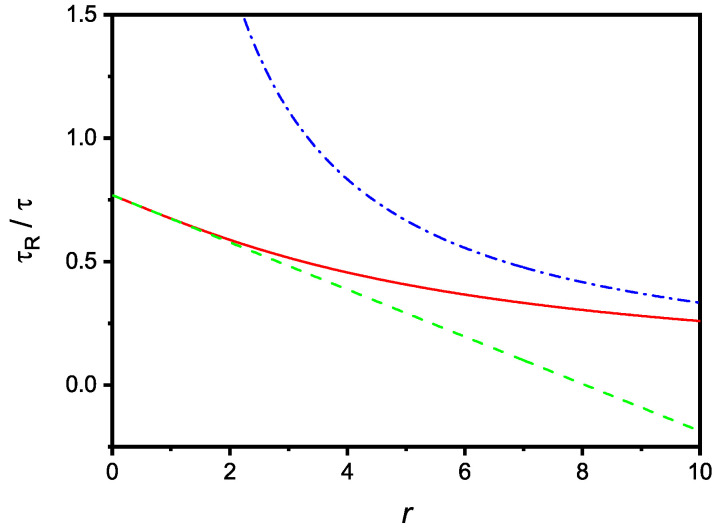
Relaxation time, in τ units, versus the dimensionless parameter *r*. The dashed and dash-dotted lines are the limiting cases of large and small *r*, respectively. The figure is drawn for u=0.3.

**Figure 4 molecules-27-07601-f004:**
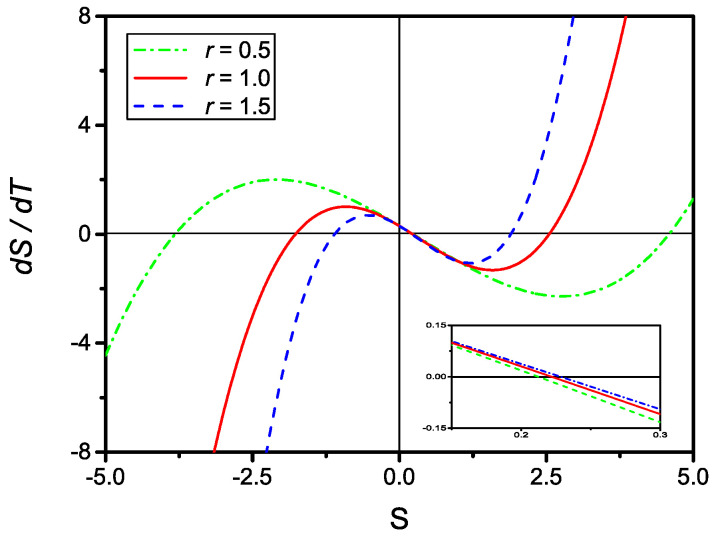
Plot of dS/dT, given by Equation (Equation 31), versus *S* for u=0.1 and r=0.5 (dash-dotted), r=1 (full), and r=1.5 (dashed). The insert shows a closeup of the three curves around S=0. Clearly, for all physically meaningful values of *r* and *u*, there are three distinct real solutions.

**Figure 5 molecules-27-07601-f005:**
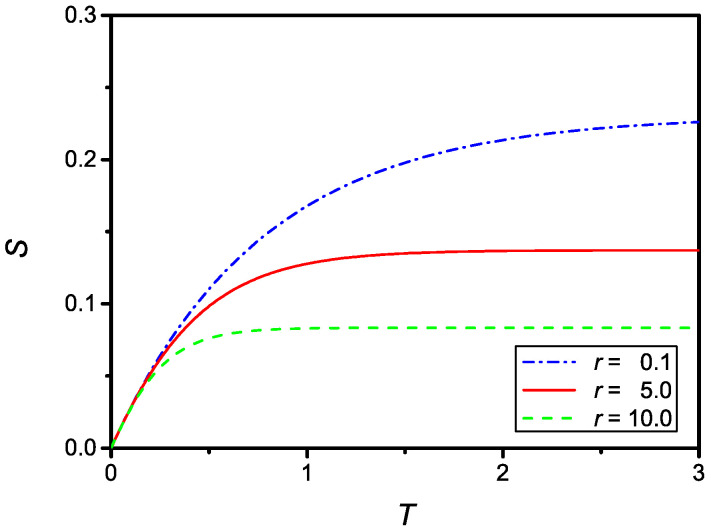
Dependence on the dimensionless surface density of adsorbed particles, S(T), versus the dimensionless time, *T*, in the case of a quadratic reduction of the adsorption coefficient. The curves were numerically obtained for u=0.3 and three values of *r*: r=0.1 (dash-dotted), r=5 (full), and r=10 (dashed).

**Figure 6 molecules-27-07601-f006:**
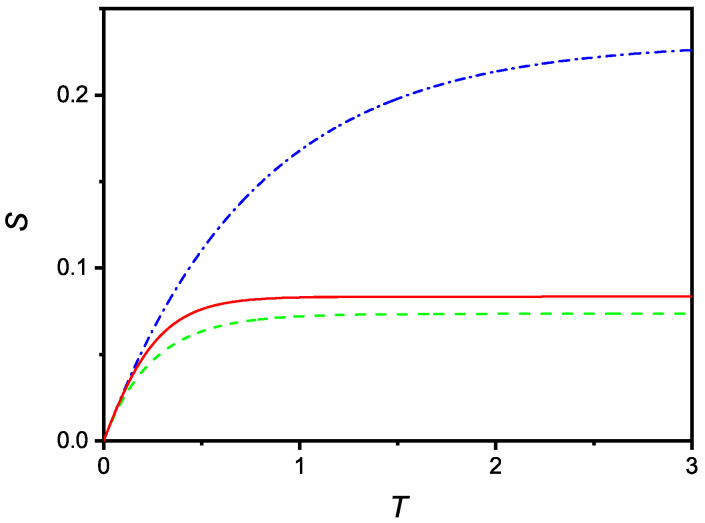
Comparison (in dimensionless units) of the time dependencies of *S* relevant to cases where saturation is absent (dash-dotted), with saturation linear in *S* (full) and saturation quadratic in *S* (dashed). The curves were drawn for u=0.3 and r=10.

**Figure 7 molecules-27-07601-f007:**
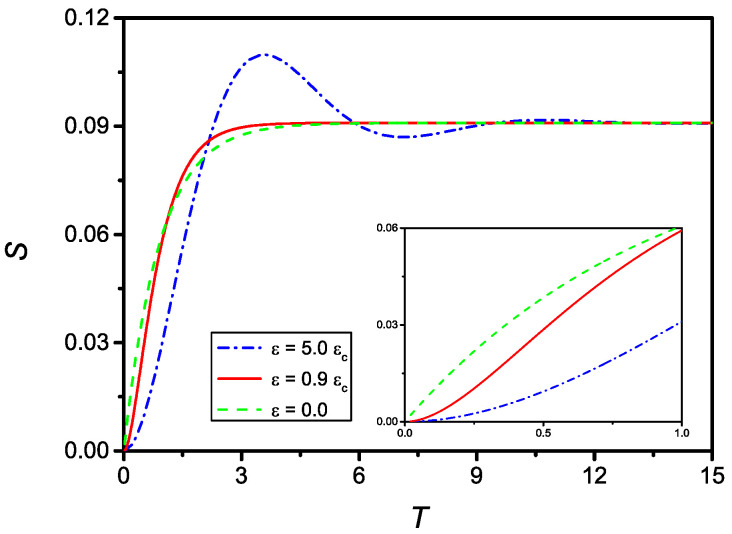
Time evolution (in dimensionless units) of *S* for $\varepsilon =5\;\varepsilon_{c}$ (dash-dotted), $\varepsilon =0.9\;\varepsilon_{c}$ (full), and δ=0 (dashed), with u=0.1. From the inset, where it is reported the evolution for small *T*, it is evident that in the case ε=0 the initial boundary condition on dS/dT is not satisfied, as a consequence from the fact that in this case the differential equation passes from the second to the first order.

**Figure 8 molecules-27-07601-f008:**
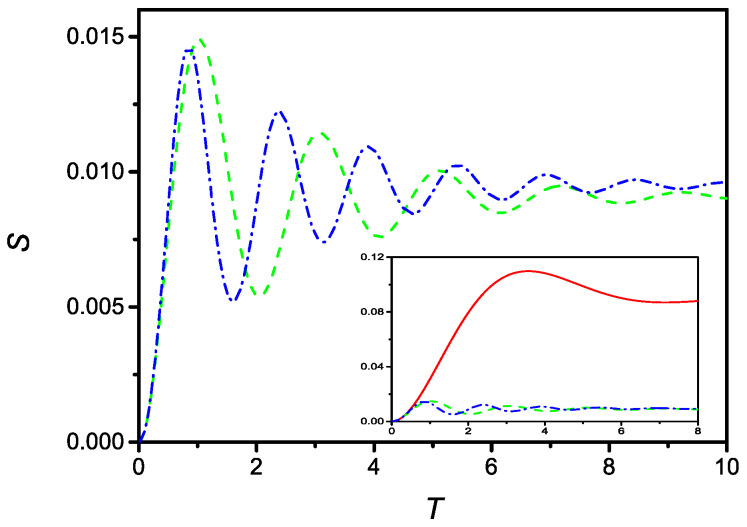
Time evolution (in dimensionless units) of the surface density of adsorbed particles in the presence of linear (dashed), and quadratic (dash-dotted), saturation effect. The curve are drawn for u=0.1, r=100 and $\varepsilon =5\;\varepsilon_{c}$. Observed in the inset, comparison between the time evolution of *S* in the absence of saturation effect (full), with linear (dashed), and quadratic (dash-dotted) saturation effect.

## Data Availability

Not applicable.

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
