# Peer review of "The Kinetics of Sorption–Desorption Phenomena: Local and Non-Local Kinetic Equations"

_molecules, 2022, doi:10.3390/molecules27217601_

Round 1
Reviewer 1 Report
In this paper, the particle density in the local kinetic equations and the adsorption-desorption process of particles on the solid-liquid interface in the non-local kinetic equations are derived by using the dynamic equation under Langmuir approximation. This work is helpful to the research of the adsorption field, but there are some problems in the description of some processes and conclusions in this paper. Therefore, I would recommend this work for publishment on Molecules after the minor revise of the following issues:
1. The article uses too many first-person "we", which is not common in scientific and technological papers. I hope to improve the sentence writing.
2. In page 1 of the article, the author says that "slab" is “one-dimensional”, but in fact it is “two-dimensional”, or the author wants to express the singleness of problem variables?
3. In page 1, line 13 of the article, "k(σ) and τ the coefficient and the desorption time, respectively." Is there a problem with the expression? It is recommended to read the full text again to correct such problems.
4. There are many formulas in the article, which may make readers feel very complicated. I suggest that the variables in the formula can be bolded.
5. The labeling suggestions following the formula can be divided according to the sections. For example, (3) is changed into (2.1.1), and (13) is changed into (2.2.1).
6. The horizontal and vertical coordinates in the illustration of the article do not give units of measurement, such as "s, m, m2", etc. Is this correct?
7. There are many ambiguous expressions in the sentences of the article. It is recommended to optimize after careful reading.
Reviewer 2 Report
1- please add a nomenclature list with all acronyms and parameters used
2- The text is not sufficiently clear, and the use of English should be improved [ for examples
Page 5: "from Eqs. (18) we get"
3- dd a paragraph in the introduction to show the author's motivation and the novelty of this work
4- please, Provide the obtained key results in the abstract and conclusion
5- Can you correlate your theoretical study with the experimental literature survey
6- why you focused on the Langumair model only, what about the other widely used models
7- Please provide a label for the Y-axis of Figure 4, and if possible please provide units for all Y- and X-axis labels for all plots
Round 2
Reviewer 2 Report
Accept in present form